# Association of the rs17574 *DPP4* Polymorphism with Premature Coronary Artery Disease in Diabetic Patients: Results from the Cohort of the GEA Mexican Study

**DOI:** 10.3390/diagnostics12071716

**Published:** 2022-07-15

**Authors:** Gilberto Vargas-Alarcón, Maria del Carmen González-Salazar, Adrian Hernández-Díaz Couder, Fausto Sánchez-Muñoz, Julian Ramírez-Bello, José Manuel Rodríguez-Pérez, Rosalinda Posadas-Sánchez

**Affiliations:** 1Department of Molecular Biology, Instituto Nacional de Cardiología Ignacio Chávez, Mexico City 14080, Mexico; gvargas63@yahoo.com (G.V.-A.); josemanuel_rodriguezperez@yahoo.com.mx (J.M.R.-P.); 2Department of Endocrinology, Instituto Nacional de Cardiología Ignacio Chávez, Mexico City 14080, Mexico; telesforo_13@yahoo.com.mx (M.d.C.G.-S.); dr.julian.ramirez.hjm@gmail.com (J.R.-B.); 3Department of Immunology, Instituto Nacional de Cardiología Ignacio Chávez, Mexico City 14080, Mexico; adrian.hernandez.diazc@hotmail.com (A.H.-D.C.); fausto22@yahoo.com (F.S.-M.)

**Keywords:** dipeptidyl peptidase-4, premature coronary artery disease, type 2 diabetes mellitus, polymorphism, DPP4 concentrations

## Abstract

Previously, it has been reported that hypoalphalipoproteinemia (HA) is associated with rs17574 *DDP4* polymorphism. Considering that in diabetic patients, HA is often present and is a risk factor for premature coronary artery disease (pCAD), the study aimed to evaluate the association of this polymorphism with pCAD in diabetic individuals. We genotyped the rs17574 polymorphism in 405 pCAD patients with T2DM, 736 without T2DM, and 852 normoglycemic individuals without pCAD and T2DM as controls. Serum DPP4 concentration was available in 818 controls, 669 pCAD without T2DM, and 339 pCAD with T2DM. The rs17574 polymorphism was associated with lower risk of pCAD (*p*_additive_ = 0.007; *p*_dominant_ = 0.003, *p*_heterozygote_ = 0.003, *p*_codominant1_ = 0.003). In pCAD with T2DM patients, DPP4 levels were lower when compared with controls (*p* < 0.001). In the whole sample, individuals with the rs17574 *GG* genotype have the lowest protein levels compared with *AG* and *AA* (*p* = 0.039) carriers. However, when the same analysis was repeated separately in all groups, a significant difference was observed in the pCAD with T2DM patients; carriers of the *GG* genotype had the lowest protein levels compared with *AG* and *AA* (*p* = 0.037) genotypes. Our results suggest that in diabetic patients, the rs17574*G* *DPP4* allele could be considered as a protective genetic marker for pCAD. DPP4 concentrations were lower in the diabetic pCAD patients, and the rs17574*GG* carriers had the lowest protein levels.

## 1. Introduction

Worldwide, cardiovascular disease (CVD) is the primary cause of death. Atherosclerosis is the leading cause of CVD. Smoking, obesity, high total cholesterol, high blood pleasure, and type two diabetes mellitus (T2DM) are some of the most significant risk factors for cardiovascular disease [1]. T2DM is a main, independent, established risk factor for coronary artery disease (CAD). T2DM patients have a two- to four-fold higher CAD risk than non-diabetic patients [2]. Moreover, in CAD patients, T2DM predicts adverse outcomes and mortality independently of other risk factors [3,4]. Hypoalphalipoproteinemia is a common feature of T2DM [5] and the most common dyslipidemia found in members of families with premature CAD (pCAD) [6]. The serin protease dipeptidyl peptidase-4 (DPP4) or CD26 is a membrane-bound exopeptidase expressed on various tissues [7]. DPP4 degrades numerous substrates by cleaving the N-terminal X-proline dipeptides from various substrates. These include regulatory peptides, chemokines, growth factors, and incretins [8]. Through a shedding process, DPP4 is released into the circulation [9]. Membrane-anchored-form and soluble DPP4 have similar enzymatic activity [10]. Due to its broad enzymatic activity, DPP4 is not only involved in the pathogenesis of T2DM but also participates in insulin resistance, fatty liver, hypertension, and oxidative stress [11,12,13,14]. DPP4 also acts as a co-stimulatory molecule that participates in T-cell biology [15]. Its assistance as a modulator of inflammation and glucose and lipid metabolism may contribute to the development of atherosclerosis [16,17]. Increased DPP4 activity and/or concentration is considered to be a manager of several metabolic abnormalities, including obesity, non-alcoholic fatty liver, T2DM, and CAD [18,19,20,21]. Enhanced expression and shedding of DPP4 from cells of diverse metabolically active tissues, including endothelium, liver, smooth muscle, and adipose tissue, may increase plasma DPP4 concentrations [18]. *DPP4* gene is located in region 2q24.3 and is highly polymorphic. Some polymorphisms have been associated with DPP4 and apolipoprotein B concentration [22,23], T2DM [23], and myocardial infarction in CAD patients [24]. In consideration, hypoalphalipoproteinemia is a common dyslipidemia found in both T2DM and premature CAD patients, and we recently reported that the rs17574 *DPP4* polymorphism was associated with a low risk of hypoalphalipoproteinemia [25]; this study aimed to investigate the association of pCAD with the rs17574 *DPP4* polymorphism and with the protein concentration in diabetic patients belonging to the GEA Mexican study.

## 2. Materials and Methods

### 2.1. Description of the Studied Population and Determination of DPP4 Polymorphisms and Concentrations

The GEA Mexican study is a cross-sectional study to evaluate the genomic basis of pCAD and determine the association of pCAD with emerging and traditional cardiovascular risk factors in a sample of Mexican-mestizo individuals from Mexico City area. The definition of pCAD was the presence of stenosis >50% demonstrated by angiography, history of percutaneous coronary intervention, coronary artery bypass grafting surgery, acute myocardial infarction, and unstable or stable or angina pectoris with at least three months before enrollment and diagnosed before 55 and 65 years of age in men and women, respectively. Controls were healthy individuals without a personal or family history of pCAD. We do not include individuals with thyroid, hepatic, chronic kidney, or malignant diseases or with current use of corticosteroids. Of the 2740 participants included in the GEA Mexican study cohort (1240 pCAD patients and 1500 healthy controls), for the present analysis, we selected 1993 individuals with complete genotype data of the rs17574 *DPP4* polymorphism: 405 were patients with pCAD and T2DM, 736 patients with pCAD without T2DM, and 852 were healthy, non-diabetic, normoglycemic, and without personal or family history of pCAD and coronary artery calcification (CAC) score equal to zero. Before their inclusion in the study, all participants provided written informed consent. The study followed the Declaration of Helsinki. The Institutional Review Board of the Instituto Nacional de Cardiología Ignacio Chávez approved the project (number 18–1082).

The evaluation and definition of anthropometric, biochemical, and clinical variables; physical activity, family, and personal medical history; smoking habits; and cardiovascular risk factors have been previously described [26,27,28]. Briefly, to calculate body mass index, we measured weight in kilograms divided by height in square meters. We used a glass fiber measuring tape to measure waist circumference in the middle point of the distance between the iliac crest and the lower side of the waist. We considered an individual as current smoking with self-reported ongoing use of cigarettes. We defined T2DM when participants reported a physician diagnosis of diabetes, treatment for glucose-lowering, or when fasting glucose was ≥ 126 mg/dL, according to the American Diabetes Association criteria.

To have a group of healthy individuals without pCAD, in all the participants of the GEA Mexican study, we decided to perform a computed tomography of the chest and abdomen and evaluate the presence of coronary artery calcium (CAC). We used the Agatston method to quantify CAC [29]. Of all the healthy individuals without pCAD with CAC scores equal to zero, we selected 852 non-diabetic individuals as a healthy control group for the present study. Physical activity was measured using the Baecke questionnaire [30]. The validated Baecke questionnaire was used to evaluate the physical activity, where the sum of the work exercise and leisure time activities was considered as the total activity.

The DPP4 concentrations were determined using a Bioplex system (R&D Systems, Minneapolis, MN, USA). DPP4 serum concentration was expressed in ng/mL. Of the 1993 individuals included in the present study, serum samples to quantify DPP4 levels were available in 1826: 818 non-diabetic normoglycemic controls, 669 pCAD non-diabetic patients, and 339 pCAD diabetic patients. We used the QIAamp DNA Blood Mini kit (QIAGEN, Hilden, Germany) for the extraction of genomic DNA from peripheral blood. The rs17574 *DPP4* polymorphism was genotyped using 5′-exonuclease TaqMan genotyping assays on an ABI Prism 7900HT Fast Real-Time PCR system (Applied Biosystems, Foster City, CA, USA).

### 2.2. Statistical Analysis 

By direct counting, we determined the frequencies of genotypes and alleles. Data are shown as median (interquartile range), frequencies, or mean ± standard deviation. For the continuous variable comparisons, we use Kruskal–Wallis or ANOVA test as appropriate. For categorical variable comparisons and to determine Hardy–Weinberg’s equilibrium, we employed the chi-square test. Kruskal–Wallis test or Mann–Whitney U test was used to evaluate the differences in DPP4 serum concentration. To test for the association of rs17574 polymorphism with pCAD in diabetic patients, we used logistic regression analysis. We adjusted each model for appropriate confounding variables. The association analyses were made under the additive, dominant, recessive, heterozygous, and co-dominant models. For the analysis of each logistic regression model, a Hosmer–Lemeshow goodness of fit test was performed. We use SPSS software v15.0 (SPSS, Chicago, IL, USA) to conduct all statistical analyses and considered a statistically significant difference when the *p*-value was <0.05. 

## 3. Results

### 3.1. General Characteristics of the Population Stratified by Group

Table 1 shows the characteristics of the study population and genotypes. Age, male percentage, BMI, waist circumference, abdominal visceral tissue (AVT), and triglycerides levels were significantly higher in pCAD patients with and without T2DM when compared with healthy subjects. As expected, the high-density lipoprotein-cholesterol (HDL-C) and apolipoprotein A1 concentration and physical activity were significantly lower in pCAD groups than in control individuals. Due to lipid-lowering treatment and the advice of changes in lifestyle, low-density lipoprotein cholesterol (LDL-C) and apolipoprotein B levels and current smoking were also lower in patients with pCAD (Table 1).

### 3.2. DPP4 Serum Concentration Stratified by rs17574 DPP4 Genotypes

DPP4 serum concentration was available for 818 non-diabetic normoglycemic controls, 669 pCAD non-diabetic patients, and 339 pCAD diabetic patients. Protein levels were higher in control individuals than in non-diabetic pCAD and diabetic pCAD patients (121 (93–155) ng/mL, 107 (76–137) ng/mL, and 93 (70–122) ng/mL, respectively, *p* < 0.001) (Table 1). The protein levels were analyzed stratifying the whole sample for rs17574 *DPP4* genotypes. Individuals with the *GG* genotype (103 (74–133) ng/mL) have the lowest protein levels followed by *AG* (108 (82–139) ng/mL) and *AA* (113 (84–146) ng/mL, *p* = 0.039) genotypes (Figure 1).

The association of rs17574 genotypes with the protein levels were evaluated separately in control, non-diabetic pCAD and diabetic pCAD individuals. In control subjects, no significant differences were observed (*p* = 0.418). For the pCAD non-diabetic group, the *GG* genotype also had the lowest DPP4 concentration (93 (63–106) ng/mL) compared with *AG* (103 (74–133) ng/mL) and *AA* genotypes (111 (77–141) ng/mL) without reaching statistically significant difference (*p* = 0.091). Similar to that observed in the pCAD non-diabetic group, in the pCAD with T2DM group, the *GG* genotype patients had the lowest protein levels (73 (59–92)) compared with carriers of the *AG* (89 (64–115) ng/mL) and *AA* genotypes (96 (72–123) ng/mL, *p* = 0.037) (Figure 2). 

### 3.3. Association of the rs17574 DPP4 Polymorphism with pCAD + T2DM

The rs17574 *DPP4* polymorphism was in Hardy–Weinberg equilibrium. To analyze the association of pCAD with the *rs17574 DPP4* polymorphism, we executed a logistic regression analysis for each inheritance model and adjusted them for sex, age, BMI, smoking habit, and LDL-C levels. Under all inheritance models tested, we found a similar distribution of the *rs17574 DPP4* polymorphism in individuals with pCAD, and in healthy controls, we did not find an association with the presence of pCAD. However, considering that in the pCAD group, the frequency of T2DM is high, we analyzed pCAD patients with and without T2DM. This analysis showed no statistically significant differences in the distribution of the rs17574 *DPP4* polymorphism in pCAD without T2DM patients and healthy controls. However, we found that the rs17574 *DPP4* polymorphism was associated with protection for pCAD in T2DM patients (pCAD + T2DM) under additive (OR = 0.68, *p* = 0.007), dominant (OR = 0.61, *p* = 0.003), heterozygote (OR = 0.61, *p* = 0.003), and co-dominant 1 (OR = 0.60, *p* = 0.003) models (Figure 3); all of them were appropriate inheritance models according to the Hosmer–Lemeshow criteria.

## 4. Discussion

To the best of our knowledge, this is the first population-based study that shows an independent association of rs17574 *DPP4* polymorphism with a more than 30% lower risk of pCAD in T2DM patients. We also found that control individuals have higher DPP4 serum concentration when compared with pCAD without T2DM and pCAD with T2DM groups. Individuals with *AA* genotype have the highest concentration of the protein in the whole sample and the pCAD with T2DM group. The lowest DPP4 concentrations were observed in individuals with *GG* genotype.

Recently, we reported that the rs17574 *G* allele was related to a 22–28% reduction of risk for the HA presence [25], dyslipidemia that often coexists with T2DM [5,31], and the most common dyslipidemia found in family members with pCAD [6]. The rs17574 polymorphism has not been previously associated with CAD. However, Aghili et al., in a sample of 875 Caucasian CAD patients, reported that the rs3788979 *DPP4* polymorphism increases the risk of myocardial infarction [24]. This polymorphism was also associated with circulating protein concentrations [24]. A similar finding was also observed in a study of 391 CAD patients and 216 individuals without CAD in the Taiwanese population, where the rs3788979 *DPP4* polymorphism increased the risk of CAD in women patients [32]. Recently, in 201 patients with CAD and T2DM, the same polymorphism was related to a smaller proportion of severe coronary artery stenosis only in women [33]. These studies support the participation of the rs3788979 *DPP4* polymorphism in the susceptibility to CAD and CAD severity in Caucasians and Chinese patients with and without T2DM. Our study did not include the determination of this polymorphism because the functional analysis we performed showed that the rs3788979 polymorphism does not have a possible functional consequence; it does not generate binding sites for transcription factors or essential sites in RNA splicing. On the other hand, we analyzed rs17574, a polymorphism with possible functional consequences. We suggest that the rs17574 polymorphism could be a genetic protector marker for pCAD in patients with T2DM in the Mexican population. These data agree with our previous finding of the protective effect of this polymorphism for the presence of HA [25]. DPP4 participates in the degradation of gastric inhibitor polypeptide (GIP) and type 1 glucagon-like peptide (GLP-1), proteins with recognized cardiovascular effects [34]. GLP-1 improves endothelial dysfunction [35] and increases coronary blood flow [36], while GIP has anti-atherosclerosis effects [37]. 

The bioinformatics analyses settled that the rs17574 *G* minor allele can change the efficiency of cutting and splicing, modifying the affinity of binding of the splicing factors SF2ASF2 and SF2ASF1 [25]. The result could be the expression of DPP4 non-functional isoforms. These proteins would be expressed to a low degree at the cellular level, and since serum DPP4 concentrations depend on cellular DPP4 shedding, these individuals would have lower levels of DPP4 in serum. On the other hand, the production of non-functional DPP4 molecules may lead to a longer half-life time of GLP-1 and GIP, which results in a prolonged-life benefit due to the cardiovascular effect of these incretins [38]. This fact could explain, at least in part, the association of the rs17574 *G* allele with a lower risk for pCAD in T2DM patients and supports the hypothesis that the rs17574*G* allele might be conceived as a potential genetic indicator for pCAD. The genome-wide expression quantitative trait loci (eQTL) dataset analysis that we performed showed that the rs17574 polymorphism affects the expression of DPP4 in the colon, lung, and visceral adipose tissue. In visceral adipose tissue and the lung, the *GG* genotype is associated with decreased expression of DPP4. It has been reported that this polymorphism is related to methylation of the gene; thus, the highest degree of methylation was observed in non-diabetic obese premenopausal women with the *GG* genotype [39] and, consequently, low concentrations of DPP4. The eQTL analysis and the Turcot report agree with our genetic results, where subjects with the *GG* genotype showed low DPP4 concentrations.

DPP4 is a novel adipokine; its expression and activity increase in obese individuals [40,41]. Fat cell volume, BMI, waist circumference, and triglycerides and leptin concentrations, all markers of obesity, are associated with plasma DPP4 concentrations [42]. *DPP4* expression is directly associated with insulin resistance (IR), especially in visceral adipose tissue, in lean and obese individuals [43]. Proinflammatory adipokines from enlarged adipocytes could regulate DPP4 release [43]. In obese T2DM and metabolic syndrome patients, increased expression and membrane shedding of DPP4 suggest that this novel adipokine could be a marker for visceral obesity, metabolic syndrome, and IR [43]. Obesity is frequently associated with the development of T2DM [44,45]; both abnormalities have in common a chronic low-grade inflammation state. DPP4 does not only modify inflammatory pathways; moreover, its expression is a marker of visceral adipose tissue (VAT) inflammation [46]. In both human and experimental obesity, when compared with lean controls, VAT dendritic cells/macrophages (1) express higher concentrations of DPP4; (2) DPP4 expression is directly associated with the degree of IR; and (3) DPP4 increases with the maturation of the dendritic cells/macrophages [46]. Considering that T2DM and CAD are abnormalities more frequently found in obese subjects and characterized by important participation of low-grade chronic inflammation, we would have expected to find a higher serum concentration of DPP4 in the pCAD with T2DM patients. On the contrary, the highest concentrations were observed in the control individual, followed by pCAD without T2DM and pCAD with T2DM (121 pg/mL vs. 107 pg/mL vs. 93 pg/mL, respectively, *p* < 0.001). Interestingly, when the DPP4 concentrations were analyzed, considering the rs17574 genotypes, the *GG* genotype showed the lowest concentration in the whole sample and in the pCAD with T2DM group. The DPP4 is released from the membrane of several metabolic active cell types, such as T lymphocytes, adipocytes, hepatocytes, smooth muscle, and endothelial cells, by a non-classical secretory mechanism of shedding [9]. The mechanism of DPP4 shedding depends on cell specificity and tissue circumstances. It is cell-type-specific and occurs due to complex coaction between diverse proteases [18]. Hypoxia increased DPP4 shedding in smooth muscle cells and adipocytes [9]. When both expression and shedding increase, the DPP4 concentration rises [18]. Several studies have established that the DPP4 activity is altered in some conditions such as obesity, non-alcoholic fatty liver, T2DM, and CAD [47,48]; however, data about the DPP4 concentrations are scarce and contradictory [18]. Gorrell et al. reported that the serum DPP4 concentrations decrease in some pathological conditions except in those where a liver injury or extensive lymphocyte proliferation is involved [49]. We previously reported decreased concentrations of DPP4 in individuals with hypoalphalipoproteinemia [25] and those with COVID-19 that required mechanical ventilation [50]. In the same way, in the present study, we reported a decrease in DPP4 concentrations in patients with pCAD and T2DM. We suggest that in those pathologies characterized by low-grade chronic inflammation, DPP4 anchored to the membrane of the immune system cells could be mainly involved in cell activation, which can reduce the degree of the DPP4 shedding by unknown mechanisms. This fact would result in a persistent state of low-grade chronic inflammation accompanied by low concentrations of DPP4. Research using animal models could help establish the molecular mechanisms associated with DPP4 involved in the genesis and progression of pCAD and T2DM.

The study’s strengths are the inclusion of a well-characterized cohort of individuals clinically, demographically, biochemically, and topographically and the evaluation of the relationship of the polymorphism with *DDP4* concentrations. However, there are some limitations: (1) we evaluated only one polymorphism located in the *DPP4* gene, (2) we did not determine the activity of DPP4, (3) patients with non-premature CAD and individuals with T2DM without pCAD were not included, and (4) it is a non-prospective cross-sectional study that does not permit establishing causality.

## 5. Conclusions

Our data establish that patients with pCAD and T2DM have the lowest DPP4 serum concentration. Individuals with T2DM carriers of the rs17574 *G* allele had more than 30% lower risk for present pCAD, and individuals carrying the *GG* genotype had the lowest concentrations of DPP4. The rs17574*G DPP4* allele could be considered as a protective genetic marker for pCAD.

## Figures and Tables

**Figure 1 diagnostics-12-01716-f001:**
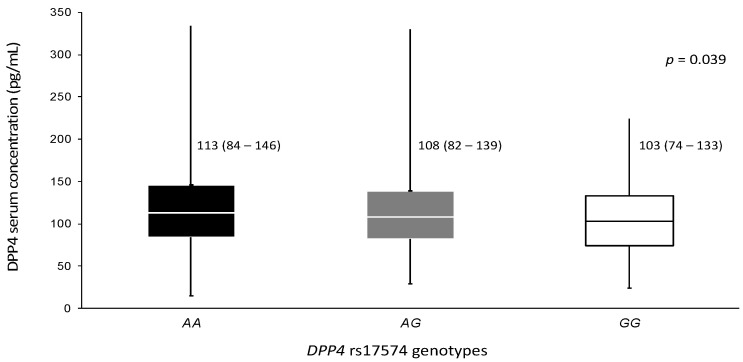
DPP4 concentration in the whole sample stratified by rs17574 genotypes. Data are presented as median (interquartile range). Differences were analyzed by the Kruskal–Wallis test.

**Figure 2 diagnostics-12-01716-f002:**
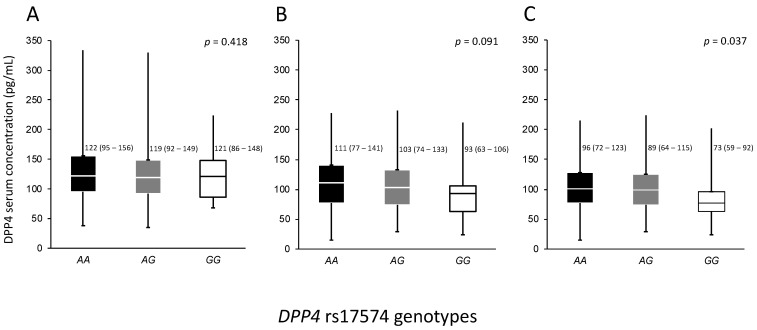
Concentration of DPP4 in the whole population stratified by *DPP4* rs17574 genotypes in each group. (**A**) Controls, (**B**) Premature CAD, and (**C**) Premature CAD + T2DM. Data are presented as median (interquartile range). CAD, coronary artery disease; T2DM, type 2 diabetes mellitus. Differences were analyzed by the Kruskal–Wallis test.

**Figure 3 diagnostics-12-01716-f003:**
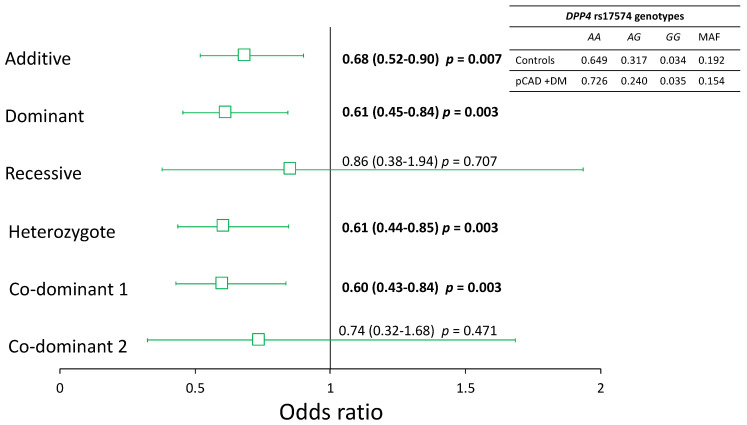
Association of the rs17574 *DPP4* polymorphism with pCAD in diabetic patients. All models were adjusted by sex, age, BMI, smoking habit, and LDL-C levels. In bold are shown the significant models.

**Table 1 diagnostics-12-01716-t001:** General characteristics of the population stratified by groups.

	Premature Coronary Artery Disease			
	No (*n* = 852)	Yes (*n* = 1141)		* *p*
	Control Group	Non-T2DM (*n* = 736)	T2DM (*n* = 405)	
Age (years)	51 ± 9	53 ± 8 ^a,b^	56 ± 8 ^a^	<0.001
Sex (% male)	40.5	85.4 ^a,b^	73.6 ^a^	<0.001
Body mass index (kg/m^2^)	27.3 (24.9–30.29)	28.1 (26.0–31.0) ^a^	28.8 (26.1–31.4) ^a^	<0.001
Waist circumference (cm)	92.2 ± 11.1	97.4 ± 10.4 ^a,b^	98.9 ± 10.5 ^a^	<0.001
Abdominal visceral tissue (cm^2^)	130 (98–172)	162 (125–208) ^a,b^	180 (137–233) ^a^	<0.001
LDL cholesterol (mg/dL)	116 (95–133)	93 (71–117) ^a,b^	86 (64–115) ^a^	<0.001
HDL cholesterol (mg/dL)	46 (37–56)	37 (31–44) ^a^	37 (32–44) ^a^	<0.001
Triglycerides (mg/dL)	138 (102–190)	159 (116–212) ^a,b^	169 (124–231) ^a^	<0.001
Apolipoprotein A1 (mg/dL)	134 (116–158)	120 (101–136) ^a^	120 (102–142) ^a^	<0.001
Apolipoprotein B (mg/dL)	92 (75 –111)	80 (65–102) ^a,b^	79 (61–102) ^a^	<0.001
DPP4 (ng/mL)	121 (93–155)	107 (76–137) ^a^	93 (70–122) ^a^	<0.001
Type 2 diabetes mellitus (%)	0	0 ^b^	100 ^a^	<0.001
Current smoking habit (%)	23.4	12.2 ^a^	10.7 ^a^	<0.001
Physical activity	7.9 (7.0–8.9)	7.6 (6.8–8.5) ^a,b^	7.4 (6.5–8.4) ^a^	<0.001
*rs17574 DPP4 genotypes*				
*AA* (%)	64.9	68.6 ^a^	72.6 ^a^	
*AG* (%)	31.7	28.1 ^a^	24.0 ^a^	0.018
GG (%)	3.4	3.5	3.5	

Data are presented as a percentage, median (interquartile range), or mean ± standard deviation. * Differences were analyzed by Kruskal–Wallis, ANOVA, or chi-square test as appropriate. ^a^ Indicates difference versus control group; ^b^ indicates difference versus premature coronary artery disease diabetes patients. T2DM, type 2 diabetes mellitus.

## Data Availability

The data presented in this study are available upon request from the corresponding author.

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
