# Peer review of "Association of the rs17574 DPP4 Polymorphism with Premature Coronary Artery Disease in Diabetic Patients: Results from the Cohort of the GEA Mexican Study"

_diagnostics, 2022, doi:10.3390/diagnostics12071716_

Round 1
Reviewer 1 Report
This is an interesting report on the association between a DPP-4 gene polymorphism and premature coronary artery disease (pCAD) in type 2 diabetic subjects.
The paper has some important criticisms, that need to be clarified before its acceptance.
First of all, neither definition nor diagnostic criteria are given about pCAD and it is very surprising, considering that this is the target population of the study
No information is available concerning the selection criteria of the study cohort: is it the entire cohort of the original study population (GEA Mexican Study) or a part? How study subjects were selected? Is there any clinical or instrumental difference between the selected study subjects (405 patients with pCAD and T2DM, 736 patients with pCAD without T2DM and 852 healthy non-diabetics, see lines 80-81) and those in which lab analysis were performed (818 non-diabetic normoglycemic controls, 669 pCAD non-diabetic patients and 339 pCAD diabetic patients, see lines 134-135)?
Results could be much more interesting if 2 other cohorts of subjects could be studied, that is subjects with non-premature CAD and T2 DM subjects without CAD. This is a limitation of the study, that should be clearly acknowledged, together with another major limitation that is its cross-sectional, non-prospective design. Overall, study limitations need to be better underscored.
Can the Authors also explain and justify why 852 healthy subjects underwent a coronary artery calcification evaluation, that is a very expensive test? Is this a screening programme or what?
Finally the Conclusion paragraph (lines 276-279) contains a summary, not the conclusion, of the study. Please rewrite it
There are also some typewriting and English mistakes (see line 17, that should be “Previously, it has been…” and lines 78-79 that should be “…1993 individuals…” ).Please check all over the paper
Author Response
1.- First of all, neither definition nor diagnostic criteria are given about pCAD and it is very surprising, considering that this is the target population of the study
Answer: As suggested by the reviewer, the inclusion criteria for pCAD were included in the material and methods section. The phrase “The definition of pCAD was the presence of stenosis >50% demonstrated by angiography, history of percutaneous coronary intervention, coronary artery by-pass grafting surgery, acute myocardial infarction, unstable or stable or angina pectoris with at least three months before enrollment and diagnosed before 55 and 65 years of age in men and women, respectively.” has been included.
2.- No information is available concerning the selection criteria of the study cohort: is it the entire cohort of the original study population (GEA Mexican Study) or a part? How study subjects were selected? Is there any clinical or instrumental difference between the selected study subjects (405 patients with pCAD and T2DM, 736 patients with pCAD without T2DM, and 852 healthy non-diabetics, see lines 80-81) and those in which lab analyses were performed (818 non-diabetic normoglycemic controls, 669 pCAD non-diabetic patients and 339 pCAD diabetic patients, see lines 134-135)?
Answer: Information about the GEA cohort and selection criteria have been included. The phrase “The GEA Mexican study is a cross-sectional study to evaluate the genomic basis of pCAD and determine the association of pCAD with emerging and traditional cardiovascular risk factors in a sample of Mexican-mestizo individuals from Mexico City area. The definition of pCAD was the presence of stenosis >50% demonstrated by angiography, history of percutaneous coronary intervention, coronary artery by-pass grafting surgery, acute myocardial infarction, unstable or stable or angina pectoris with at least three months before enrollment and diagnosed before 55 and 65 years of age in men and women, respectively. Controls were healthy individuals without a personal or family history of pCAD. We do not include individuals with thyroid, hepatic, chronic kidney, or malignant diseases or with current use of corticosteroids. Of the 2740 participants included in the GEA Mexican study cohort (1240 pCAD patients and 1500 healthy controls), for the present analysis, we selected 1993 individuals with complete genotype data or the rs17574 DPP4 polymorphism: 405 were patients with pCAD and T2DM, 736 patients with pCAD without T2DM, and 852 were healthy non-diabetic normoglycemic, without personal or family history of pCAD and coronary artery calcification (CAC) score equal to zero. Before their inclusion in the study, all participants provided written informed consent. The study followed the Declaration of Helsinki. The Institutional Review Board of the Instituto Nacional de Cardiología Ignacio Chávez approved the project (number 18-1082).” has been included in the material and methods section.
The number of individuals included in the analysis of the DPP4 concentrations was different from that in the genetic analysis. In order to clarify this point, the phrase “Of the 1993 individuals included in the present study, serum samples to quantify DPP4 levels were available in 1826: 818 non-diabetic normoglycemic controls, 669 pCAD non-diabetic patients, and 339 pCAD diabetic patients.” has been included in the material and methods section.
3.- Results could be much more interesting if 2 other cohorts of subjects could be studied, that is subjects with non-premature CAD and T2 DM subjects without CAD. This is a limitation of the study, that should be clearly acknowledged, together with another major limitation that is its cross-sectional, non-prospective design. Overall, study limitations need to be better underscored.
Answer: As the reviewer comments, the lack of individuals with non-premature CAD and individuals with DM without pCAD in the study can be considered a limitation along with the fact that it is a non-prospective cross-sectional study. These limitations have been added in the discussion section. The sentence “3) patients with non-premature CAD and individuals with T2DM without pCAD were not included, 4) is a non-prospective cross-sectional study that does not permit establishing causality.” has been added.
4.- Can the Authors also explain and justify why 852 healthy subjects underwent a coronary artery calcification evaluation, that is a very expensive test? Is this a screening programme or what?
Answer: As is commented in the manuscript, the individuals included in the present study belong to the GEA study. This is an institutional study designed to determine the relationship between traditional and emerging risk factors and genetic bases of pCAD in an adult Mexican population. In order to have a group of healthy individuals without pCAD, we decide to perform a computed tomography of the chest and abdomen, a noninvasive method and evaluate the presence of coronary artery calcium. In order to clarify this point, the phrase “To have a group of healthy individuals without pCAD, in all the participants of the GEA Mexican study, we decided to perform a computed tomography of the chest and abdomen and evaluate the presence of coronary artery calcium (CAC). We used the Agatston method to quantify CAC [23]. Of all the healthy individuals without pCAD with CAC scores equal to zero, we select 852 non-diabetic individuals as a healthy control group for the present study.” has been added in the material and methods section.
5.- Finally the Conclusion paragraph (lines 276-279) contains a summary, not the conclusion, of the study. Please rewrite it.
Answer: As suggested by the reviewer, the conclusion has been rewritten. The new conclusion is “Our data establish that patients with pCAD and T2DM have the lowest DPP4 serum concentration. Individuals with T2DM carriers of the rs17574 G allele have more than 30% lower risk for present pCAD, and individuals carrying the GG genotype had the lowest concentrations of DPP4. The rs17574G DPP4 allele could be considered as a protective genetic marker for pCAD.”
There are also some typewriting and English mistakes (see line 17, that should be “Previously, it has been…” and lines 78-79 that should be “…1993 individuals…” ). Please check all over the paper
Answer: The typewriting and English mistakes have been corrected. The text has been checked.
Reviewer 2 Report
In the article ‘Association of the rs17574 DPP4 polymorphism with premature coronary artery disease in diabetic patients. Results from the cohort of the GEA Mexican study’, the author tries to study the relationship between rs17574 DPP4 polymorphism and premature coronary artery disease in diabetic patients. It is a valuable study. However, there are some shortcomings and questions.
1)In the line 154-156: Our data suggest that the rs17574G DPP4 allele could be considered a protective genetic marker for pCAD in diabetic subjects, It should be: Our data suggests that the rs17574G DPP4 allele could be considered as a protective genetic marker for pCAD in diabetic subjects.
2)The English language should be improved overall.
3) All the figures are needed to improve the figure legends.
4) The molecular mechanism is needed to explore further, especially in the animal model, although it is beyond the scope of this article.
Author Response
1)In the line 154-156: Our data suggest that the rs17574G DPP4 allele could be considered a protective genetic marker for pCAD in diabetic subjects, It should be: Our data suggests that the rs17574G DPP4 allele could be considered as a protective genetic marker for pCAD in diabetic subjects.
Answer: As suggested by the reviewer, the phrase has been corrected.
New phrase: “Our results suggest that in diabetic patients, the rs17574G DPP4 allele could be considered as a protective genetic marker for pCAD.”
2)The English language should be improved overall.
Answer: The English language has been improved
3) All the figures are needed to improve the figure legends.
Answer: The figure legends have been improved
4) The molecular mechanism is needed to explore further, especially in the animal model, although it is beyond the scope of this article.
Answer: As commented by the reviewer, the animal models permit to the exploration of the molecular mechanisms related to the diseases, however, this is beyond the scope of our article.
This point is considered in the discussion section and the phrase "Research using animal models could help establish the molecular mechanisms associated with DPP4 involved in the genesis and progression of pCAD and T2DM.”
Reviewer 3 Report
It is well-conducted research and well-written manuscript, regarding the ischemic coronary risk in diabetic patients with rs17574 DPP4 polymorphism, in a large cohort of patients belonging to the GEA Mexican Study.
Author Response
It is well-conducted research and well-written manuscript, regarding the ischemic coronary risk in diabetic patients with rs17574 DPP4 polymorphism, in a large cohort of patients belonging to the GEA Mexican Study.
Answer: Thank you very much for your positive comments.
Round 2
Reviewer 1 Report
No further comments
Author Response
Thank you very much for your positive comments. A minor spell check has been done.